# Concurrent oxygen reduction and water oxidation at high ionic strength for scalable electrosynthesis of hydrogen peroxide

Changmin Kim[1], Sung O Park[2], Sang Kyu Kwak [2,3], Zhenhai Xia[1], Guntae Kim [4] ✉ & Liming Dai [1] ✉

Electrosynthesis of hydrogen peroxide via selective two-electron transfer oxygen reduction or water oxidation reactions offers a cleaner, cost-effective alternative to anthraquinone processes. However, it remains a challenge to achieve high Faradaic efficiencies at elevated current densities. Herein, we report that oxygen-deficient $Pr_{1.0}Sr_{1.0}Fe_{0.75}Zn_{0.25}O_{4-\delta}$ perovskite oxides rich of oxygen vacancies can favorably bind the reaction intermediates to facilitate selective and efficient two-electron transfer pathways. These oxides exhibited superior Faradic efficiencies (~99%) for oxygen reduction over a wide potential range (0.05 to 0.45 V *versus* reversible hydrogen electrode) and current densities surpassing 50 mA cm$^{-2}$ under high ionic strengths. We further found that the oxides perform a high selectivity (~80%) for two-electron transfer water oxidation reaction at a low overpotential (0.39 V). Lastly, we devised a membrane-free electrolyser employing bifunctional electrocatalysts, achieving a record-high Faradaic efficiency of 163.0% at 2.10 V and 50 mA cm$^{-2}$. This marks the first report of the concurrent oxygen reduction and water oxidation catalysed by efficient bifunctional oxides in a novel membrane-free electrolyser for scalable hydrogen peroxide electrosynthesis.

Hydrogen peroxide ($H_2O_2$) is one of the most important chemicals for chemical synthesis, water treatment, paper manufacture, and energy storage[1,2]. The current anthraquinone process for industrial production of $H_2O_2$ requires expensive catalysts and a considerable amount of hydrogen gas with intensive energy consumption[3]. The instable nature of $H_2O_2$ (e.g., self-explosive under a concentrated condition) also poses safety issues for transportation and storage[4]. Therefore, electrochemical $H_2O_2$ synthesis either from oxygen reduction reaction (ORR) or water oxidation reaction (WOR)[1,5,6] powered by renewable energy has recently become an attractive option for cost-effective, efficient, and on-site clean production of $H_2O_2$.

As is well-known, the oxygen reduction reaction (ORR) can proceed either by a two-electron (2e-) pathway with the formation of $H_2O_2$ (in acidic medium) or $HO_2^-$ (in alkaline medium) as the intermediate species or by a four-electron (4e-) process to reduce $O_2$ into $H_2O$ (in acidic medium) or $OH^-$ (in alkaline medium)[1,4]. On the other hand, $H_2O_2$ can also be formed by water oxidation reaction (WOR) through a 2e- pathway while 4e-WOR will lead to oxygen evolution reaction (OER)[5–7]. For selective $H_2O_2$ electrosynthesis, therefore, considerable effort has been made to develop novel catalyst materials capable of suppressing the competitive 4e-ORR or 4e-OER reactions[3–6].

During oxygen reduction, $OO^* \rightarrow HOO^* \rightarrow HOOH$ or $OO^* \rightarrow HOO^* \rightarrow O^* \rightarrow HO^*$), the dissociation of $OH^-$ from $HOO^*$ intermediate determines whether the overall reaction proceeds through a 2 or 4 electron pathway[6–8]. Thus, many studies have been performed to find the optimum Gibbs free energy ($\Delta G$)[9–15] for binding between the

[1]Australian Carbon Materials Centre (A-CMC), School of Chemical Engineering, University of New South Wales, Sydney, NSW 2052, Australia. [2]Department of Energy Engineering, Ulsan National Institute of Science and Technology (UNIST), Ulsan 44919, South Korea. [3]Department of Chemical and Biological Engineering, Korea University, Seoul 02841, South Korea. [4]Key Laboratory of Interfacial Physics and Technology, Shanghai Institute of Applied Physics, Chinese Academy of Sciences, Shanghai 201800, China. ✉ e-mail: gtkim@sinap.ac.cn; l.dai@unsw.edu.au

catalytic surface and oxygenic species. It was found that a moderate $\Delta G_{HOO^*}$ of around 4.2 eV is neither too strong to properly release oxygenic species to complete the reaction cycle nor too weak to prevent the dissociation of OH⁻. These findings have been used to guide the design and development of new metal-based 2e-ORR catalysts, such as Pt-Hg[6], Pd-Hg[13], and Au-Pt-Ni[15] nanoparticles with the optimized binding energy (~4.2 eV) and high Faradaic efficiencies (FEs, over 95%) around the onset potential (~0.5 V *vs*. RHE). For high current-density and/or industrial scale $H_2O_2$ production, however, reduced FEs are often obtained at high overpotentials ($\eta$) or high ionic strength conditions[6,14]. To realize a large-scale efficient $H_2O_2$ electrosynthesis, therefore, it is important to achieve high FEs particularly at a high current density, high ionic strength (e.g., 0.5–2.0 M electrolytes), and/or high $\eta$ (e.g., 0.0–0.2 V *vs*. RHE).

In this study, we developed a new class of oxygen-deficient $Pr_{1.0}Sr_{1.0}Fe_{0.75}Zn_{0.25}O_{4-\delta}$ (D-PSFZ) perovskite oxides with optimized superior FEs (over 99%) over the entire oxygen reduction potential regions (i.e., 0.05–0.45 V *vs*. RHE) even in electrolytes with high ionic strengths (i.e., 0.5–2.0 M $KHCO_3$). These perovskite oxides possess a unique structural tuneability at the atomic level ($ABO_3$, having the A site with rare earth metal ions and the B site with transition metal ions)[16], and allow for the control of electrochemical activities and fundamental understanding of the structure-performance relationship. Specifically, we observed that the formation of oxygen vacancies ($Ov$) within D-PSFZ activated reactive sites (i.e., Zn, and Fe cations) can reinforce the binding of HOO* intermediate (i.e., $\Delta G_{HOO^*}$ negative shift) to prevent the dissociation of HOO* into O* and OH⁻, and hence eliminating the 4e transfer pathway. In contrast, its counterpart, $Pr_{1.0}Sr_{1.0}Fe_{1.0}O_{4-\delta}$ (PSF), with no $Ov$ showed a severe drop in the FE at high $\eta$ regions and high solution concentrations.

Our findings indicate that the stronger coordination between the weak hydration shell cations (e.g., K⁺; hydration number, K⁺ <Na⁺ <Li⁺)[17,18] and the adsorbed HOO* intermediate can impede a selective $H_2O_2$ production pathway (*vide infra*). Thus, the binding free energy of O* (i.e., $\Delta G_{O^*}$) could be adopted as an additional parameter, along with $\Delta G_{HOO^*}$, to determine the selective $H_2O_2$ production pathway. Indeed, we found that $\Delta G_{O^*}$ is a more decisive parameter to indicate selective FEs at high $\eta$ regions while the general $\Delta G_{HOO^*}$ parameter can be used to predict the thermodynamically lowest potential to derive 2e-ORR. We also found that D-PSFZ displayed a notably high 2e-WOR selectivity of 80.0% at a significantly low potential of 2.15 V *vs*. RHE whereas the recently-reported 2e-WOR catalysts (e.g., ZnO and $BiVO_4$) exhibited similar FEs around 3.0 V – the higher potential could cause possible oxide corrosions[19,20]. Furthermore, these bifunctionality of D-PSFZ enabled us to develop the first membrane-free electrolyser with the D-PSFZ (2e-ORR) ‖ D-PSFZ (2e-WOR) configuration for overall $H_2O_2$ synthesis from 2e-ORR at the cathode and 2e-WOR at the anode with a theoretical FE of 200% – outperformed all the conversational half cells with either 2e-ORR or 2e-WOR only to produce $H_2O_2$.

Finally, we discovered that the formation of $Ov$ can favorably shift $\Delta G_{HOO^*}$ and $\Delta G_{O^*}$ to achieve highly selective ORR and WOR bifunctional activities for efficient overall $H_2O_2$ electrosynthesis (FE = 163.0% at 2.10 V & 50 mA cm⁻²) in the D-PSFZ‖D-PSFZ electrolyser, even under locally cation-concentrated conditions where the selective $H_2O_2$ production pathways are significantly facilitated. As far as we are aware, D-PSFZ is the first perovskite oxide catalyst with highly selective ORR and WOR bifunctional activities for efficient overall $H_2O_2$ electrosynthesis in a membrane-free cell, representing a breakthrough toward cost-effective, efficient, and clean electrochemical production of $H_2O_2$ of practical significance.

## Results and discussion
### Structural characterization
Oxygen nonstoichiometry ($\delta$) of Ruddlesden-popper (RP) phased perovskite oxides can be both oxygen-deficient or oxygen-excessive structures depending upon their cation compositions[21,22]. The substitution of an aliovalent cation on A- or B-site can lead to an increase of oxygen deficiency in RP oxides[21,22]. In this study, oxygen-deficient $Pr_{1.0}Sr_{1.0}Fe_{0.75}Zn_{0.25}O_{4-\delta}$ (D-PSFZ) perovskite oxides were prepared by doping with divalent ions (i.e., $Zn^{2+}$) at the B-site of $Pr_{1.0}Sr_{1.0}Fe_{1.0}O_{4-\delta}$ (PSF) oxides. The optimum ratio of Fe to Zn (i.e., 0.75:0.25) was determined by scanning the structural stability (Supplementary Fig. 1). The amount of oxygen vacancies at D-PSFZ and PSF have been determined as $\delta = 0.14$–0.15 and 0.0, respectively, via iodometry titration (see, Method).

Figure 1a, b presents bright-field transmission electron microscopy (TEM) image and high-angle annular dark-field (HAADF) scanning TEM (STEM) image and energy-dispersive spectroscopy (EDS) elemental mapping for D-PSFZ, indicating homogenous distributions for Pr, Sr, Fe, Zn, and O in D-PSFZ. Additional analysis on the morphology of D-PSFZ was performed on a scanning electron microscopy (SEM) (Supplementary Fig. 2). From the STEM–EDS mapping and the point spectrum analysis (Supplementary Fig. 3), the stoichiometric atomic ratios for all the lattice elements (i.e., Pr, Sr, Fe, Zn, and O) in D-PSFZ were identified (Supplementary Table 1). Figure 1c presents a high-resolution TEM (HR-TEM) image and the corresponding fast Fourier transform (FFT) pattern (inset) for D-PSFZ along the <100> zone axis, revealing its high crystallinity. To identify the crystalline structure of D-PSFZ oxide, we performed atomic-scale HAADF STEM and X-ray diffraction (XRD). As shown in Fig. 1d, D-PSFZ exhibits an RP perovskite structure with an obvious contrast between A- and B-site cations and the lattice spaces of 0.386 and 1.278 nm along (100) and (001) planes, respectively. Also, the Rietveld refinement analysis of D-PSFZ (Fig. 1e) confirms that it has the RP perovskite structure with a similar lattice spacing as that shown by the HAADF STEM image (Fig. 1d), without any noticeable impurity. Lattice parameters of PSF from the Rietveld refinement and the XRD pattern are shown in Supplementary Table 2 and Supplementary Fig. 4, respectively.

To identify the surface and in-depth chemical states for PSF and D-PSFZ, we further performed X-ray photoelectron spectroscopy (XPS) (Figs. 1f, g, Supplementary Figs. 5–8). As shown in Supplementary Figs. 5–8, the XPS survey spectra, along with the corresponding high-resolution Pr 3$d$ spectra of the deconvoluted peaks located near 928 eV, show similar patterns with similar $Pr^{3+}$:$Pr^{4+}$ ratios for PSF ($Pr^{3+}$:$Pr^{4+}$ = 83.4:16.6) and D-PSFZ ($Pr^{3+}$:$Pr^{4+}$ = 82.2:17.8), indicating that the partial substitution of Zn did not change the chemical state of the A-site. For Fe cation of the B-site, the deconvoluted peaks of Fe 2$p$ around 710 eV exhibit similar $Fe^{2+}$:$Fe^{3+}$ ratios for PSF ($Fe^{2+}$:$Fe^{3+}$ = 71.3:28.7) and D-PSFZ ($Fe^{2+}$:$Fe^{3+}$ = 66.4:33.6) (Supplementary Figs. 5d and 6d). However, the high-resolution XPS O 1$s$ spectra for PSF and D-PSFZ reveal significant differences (Fig. 1f, g). For both samples, the deconvoluted peaks[23] include lattice oxygen (~528.6 eV, $Ol$), oxygen vacancies (~530.0 eV, $Ov$), adsorbed oxygen species (~531.09 eV, $Oads$), and adsorbed water (~532.4 eV, $Ow$). However, the area ratio of $Ov$/$Ol$ for D-PSFZ ($Ov$/$Ol$ = 0.194) is much higher than that of PSF ($Ov$/$Ol$ = 0.032) at the surface, as shown in Fig. 1f, g and summarized in Supplementary Tables 3 and 4. These results confirm that the substitution with divalent ions (i.e., $Zn^{2+}$) can introduce oxygen-deficiencies, as observed by the iodometric method (*vide supra*). The XPS depth-profiles further confirm that the $Ov$ distributes abundantly on the surface and converges slowly at the lattice level (Fig. 1g), while PSF presents no deconvoluted $Ov$ band in the inner lattice (Fig. 1f). Brunauer-Emmett-Teller (BET) surface area and pore size distribution profiles by Barrett–Joyner–Halenda (BJH) method are shown in Supplementary Fig. 9.

### 2e-ORR activities
Electrochemical oxygen reduction activities of D-PSFZ were investigated by a rotating ring disk electrode (RRDE) method. In this work, the ORR activities were identified in a quasi-neutral media (e.g., 0.1 M $KHCO_3$) to keep $H_2O_2$ stable, instead of using an alkaline media that

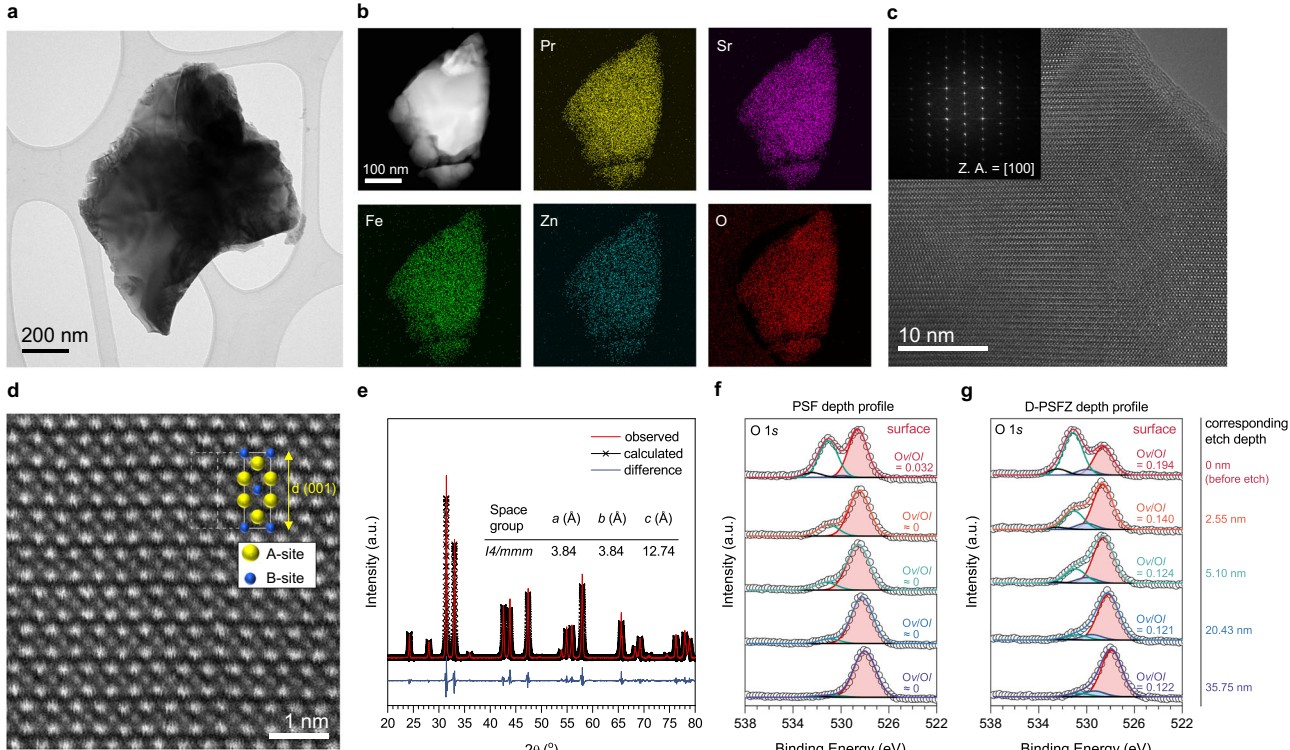

**Fig. 1 | Morphological and structural characterization of D-PSFZ. a** Bright-field TEM image of D-PSFZ. **b** STEM-HAADF image and corresponding STEM-EDS elemental mapping of D-PSFZ, presenting uniform distribution of Pr (yellow), Sr (violet), Fe (green), Zn (cyan), and O (red). **c** HR-TEM image of D-PSFZ and SAED patterns as an inset. **d** Atomic-scale HAADF-STEM image of D-PSFZ along <100> zone axis. **e** XRD spectra and the Rietveld refinement profile of D-PSFZ. **f** XPS depth profile analysis in O 1s region for PSF and **g** D-PSFZ and the corresponding Ov/Ol ratio upon the depth as the footnote.

readily decomposes $H_2O_2$ due to its high acidity (i.e., $pKa(H_2O_2) = 11.75$)[24]. Figure 2a shows the RRDE polarization curves obtained in $O_2$-saturated 0.1 M $KHCO_3$ for D-PSFZ. As can be seen, D-PSFZ exhibited an onset potential near 0.55 V (*vs.* RHE, calibration profiles are shown in Supplementary Fig. 10) and a diffusion-controlled limiting current density around 3.0 mA cm$^{-2}$, which is half the theoretical mass transport limiting current for the 4e-ORR usually found on Pt/C[14,25], indicating a highly selective 2e-ORR pathway. To accurately determine the $H_2O_2$ yield, we measured the ring current using RRDE under the $N_2$-saturated condition and subtracted it as the background current, as shown in Supplementary Fig. 11. It was found that PSF exhibited a high $H_2O_2$ selectivity close to 87% at 0.30 V, and the selectivity decreased with increasing the overpotential (e.g., 59% at 0.05 V) (Supplementary Fig. 12a).

By contrast, D-PSFZ displayed a superior $H_2O_2$ selectivity over 95% at various potentials, ranging from 0.20 to 0.45 V, with the electron transfer number close to ~2.0 (Fig. 2a). From the ORR polarization profiles under 400–2500 rpm for D-PSFZ and Pt/C shown in Supplementary Fig. 13, the Koutecky–Levich plots were obtained (Fig. 2b and Supplementary Fig. 13), giving a Levich constant around 9.20–9.38 mA$^{-1}$ cm$^2$ s$^{-1/2}$ for D-PSFZ and around 4.61–4.70 mA$^{-1}$ cm$^2$ s$^{-1/2}$ for Pt/C. These results indicate that D-PSFZ follows a 2e-transfer pathway for ORR whilst Pt/C follows a selective 4e-ORR pathway (see also, Fig. 2a and Supplementary Fig. 14). PSF exhibited a Levich constant of 8.01 mA$^{-1}$ cm$^2$ s$^{-1/2}$, which is slightly smaller than that observed for D-PSFZ (i.e., 9.20–9.38 mA$^{-1}$ cm$^2$ s$^{-1/2}$), confirming a moderately selective 2e-ORR pathway for PSF (Supplementary Figs. 12a and 13c, d).

To systematically investigate the 2e-ORR selectivity for D-PSFZ, we proceeded the RRDE testing at chronoamperometric (CA) conditions from 0.45 to 0.15 V and subtracted the background current profiles by measuring the CA profiles under both the $O_2$- and $N_2$-saturated conditions (Supplementary Fig. 15). As shown in Fig. 2c, D-PSFZ shows a

superior $H_2O_2$ selectivity over 90% for the potential range over 0.45 to 0.15 V - nearly 98% of the selectivity was found from 0.35 to 0.25 V. It is worth noting that D-PSFZ follows a highly selective 2e-ORR pathway over the broad potential range, indicating its potential for high-current-density electrolysis. Based on these findings, we envision that the formation of lattice oxygen deficiencies in D-PSFZ promotes its stable binding with the reaction intermediate (i.e., HOO*) to tune the reaction activity of the active sites, as we shall see in more details later.

To demonstrate the potential of D-PSFZ for practical applications, we deposited the D-PSFZ oxide onto a piece of carbon paper and utilized it as a working electrode in a typical H-cell configuration. Figure 3a displays the CV polarization profiles for D-PSFZ (the lower part) measured in solutions with various $KHCO_3$ concentrations, and the $H_2O_2$ selectivities (i.e., FEs; the upper part) at potentials over the entire ORR range from 0.45 V (near onset potential) to 0.05 V (high overvoltage region). We used a test paper stripping, a permanganate titration method, or ultraviolet-visible (UV-vis) spectroscopy to determine the $H_2O_2$ yields, as shown in Method and Supplementary Table 5-29. In particular, the CA profiles were measured over the entire potential range (i.e., 0.45–0.05 V, every 0.05 or 0.1 volts), and we also used the titration method to determine the $H_2O_2$ yield based on the transferred charge during the CA measurements (Supplementary Fig. 16 and Supplementary Tables 5–9). Once again, it was found that D-PSFZ showed highly efficient 2e-ORR activities with the $H_2O_2$ selectivity over 99.5% across the entire potential range from 0.05 to 0.45 V *vs.* RHE in $O_2$-sat'd 0.1 M $KHCO_3$ (black line) (Fig. 3a and Supplementary Fig. 16a). As the electrolyte concentration increased from 0.1 to 0.5 M $KHCO_3$ (red line), much higher current densities (e.g., 38 mA cm$^{-2}$ at 0.05 V, Fig. 3a) were obtained due to the decreased solution resistance with increasing $KHCO_3$ concentration (Supplementary Fig. 17)[26].

As can be seen in Fig. 3a, b, D-PSFZ shows FEs >99% over the entire ORR range with a $H_2O_2$ yield rate of 675 μmol h$^{-1}$ at 0.05 V in 0.5 M

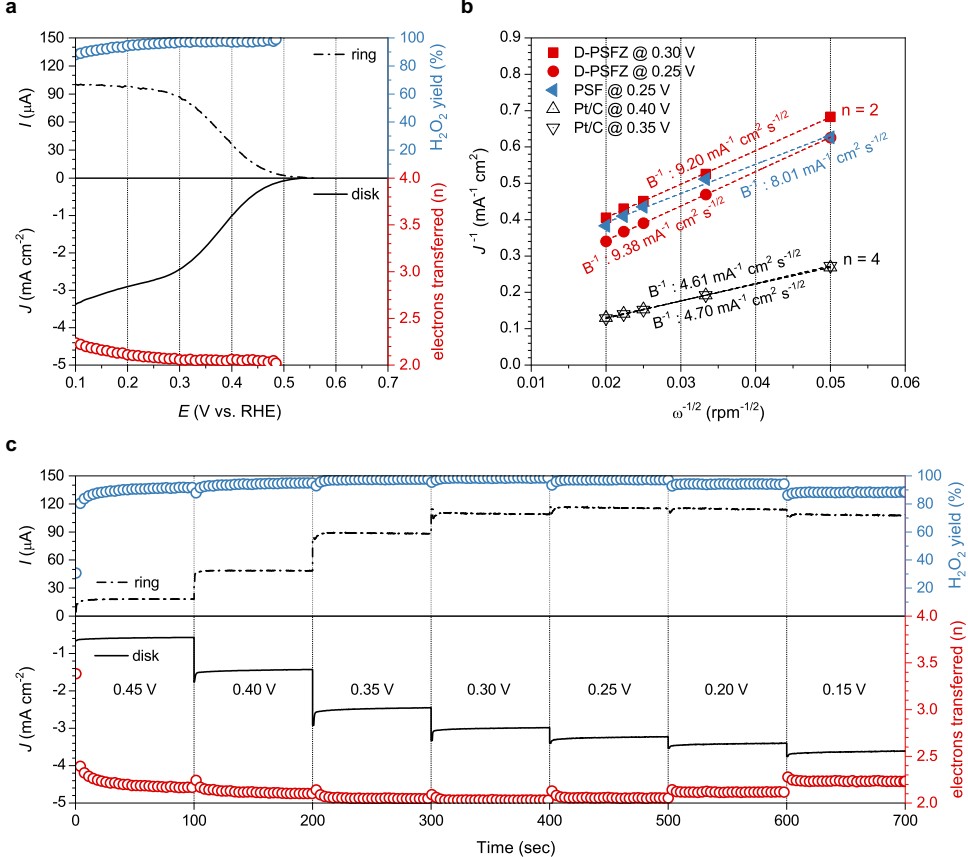

**Fig. 2 | RRDE analysis for D-PSFZ conducted in O$_2$-saturated 0.1 M KHCO$_3$.** The potential was referenced to the RHE as experimentally determined from the calibration profile. *iR* drops were not compensated for all data set in this work. **a** RRDE polarization profiles for oxygen reduction reaction. **b** The Koutechy–Levich slope analysis for D-PSFZ, PSF, and Pt/C. **c** The CA profiles of D-PSFZ at the potential range from 0.45 to 0.15 V.

KHCO$_3$. At higher concentrations of 1.0 M (blue line) and 2.0 M (green line) KHCO$_3$, similar FEs close to 99% were obtained at the low-middle overpotential range from 0.15 V to 0.45 V (Fig. 3a). It is worth noting that the FEs are as high as 99% even at high operating current densities up to 60 mA cm$^{-2}$ in 1.0 or 2.0 M KHCO$_3$ over 0.15–0.45 V (Fig. 3a). Especially, under the high $\eta$ of 0.05 V in the high ionic strength (i.e., 2.0 M KHCO$_3$), the H$_2$O$_2$ yield rate was determined to be 1035 μmol h$^{-1}$ with a FE of 95.6% (Fig. 3b). These results indicate once again that the oxygen-deficient oxide, D-PSFZ, follows a highly selective 2e-ORR pathway, and that the electrolyte concentration plays an important role in regulating the H$_2$O$_2$ yield. At the high KHCO$_3$ concentration and high overpotential region (i.e., 0.10–0.05 V), however, FEs tend to decrease slightly to about 95% (Fig. 3a), indicating that the HOO* intermediates proceed partially to dissociation into O* and OH$^-$ intermediates following a 4e-ORR pathway.

In saturated KHCO$_3$ (~3.4 M) (orange line), a decrease in FEs down to 80% over the entire ORR potential range was observed (Fig. 3a). In the saturated KHCO$_3$, therefore, the dissociation of the reaction intermediate, HOO*, into O* and OH$^-$ via a 4e-ORR pathway became more significant. To identify this susceptible dissociation of HOO* into O* and OH$^-$ under the high ionic strength conditions, we further examined the 2e-ORR activities for PSF and oxidized acetylene black (AB). Supplementary Fig. 18 presents the visualized FEs as a function of potentials and KHCO$_3$ concentrations for PSF and AB. As shown in Supplementary Fig. 18b, the gradual decrease of FEs (from 99.2 to 73.4%) was observed when $\eta$ increased from 0.45 to 0.05 V at 0.1 M KHCO$_3$. Likewise, FEs reduced (from 99.2 to 72.1%) when the ionic strength increased (from 0.1 to 2.0 M) at 0.45 V. Also, a similar trend of FEs was observed for AB (Supplementary Fig. 18a). These results

demonstrate that both $\eta$ and ionic strength can influence HOO* dissociation that is decisive for the FEs. In this regard, we envision that a locally cation-concentrated layer within a cationic double-layer (i.e., Helmholtz plane) formed by high $\eta$ or ionic strength conditions can impede the stable binding of HOO* intermediate. Thus, the cationic effect at high $\eta$ regions was further investigated using 0.1 M NaHCO$_3$ for PSF. When PSF was tested in 0.1 M NaHCO$_3$, it exhibited stable FEs of about 80% in a wide potential range over 0.50–0.05 V (Supplementary Fig. 12b). Whereas in KHCO$_3$, a significant decrease in FE from 87.4 to 58.9% was observed when the $\eta$ increased from 0.30 to 0.05 V (Supplementary Fig. 12a). This clearly indicates that the stronger coordination between the weak hydration shell cation (i.e., K$^+$) and the adsorbed HOO* intermediate can facilitate its dissociation into O* and OH$^-$, leading to the partial 4e-ORR pathway. In view of the degree of hydration (hydration number, K$^+$ <Na$^+$ <Li$^+$)[17,18], it would be reasonable to use NaHCO$_3$ solution for a selective 2e-ORR pathway because highly hydrated cations can have less coordination with the reaction intermediate at the surface of electrode. However, the lower ionic conductivity and solubility of NaHCO$_3$ in water[26,27] compared to KHCO$_3$ can lead to a lower working current density (Supplementary Fig. 12) and energy efficiency, eventually leading to more favorable and selective H$_2$O$_2$ synthesis in KHCO$_3$.

Figure 3c shows the color-mapped contour plots of the FEs for D-PSFZ as a function of the applied potentials and solution KHCO$_3$ concentrations, along with the FEs for recently-reported 2e-ORR catalysts[6,9,12,28,29]. As can be seen, D-PSFZ follows the highly selective 2e-ORR pathway at a wide potential range over 0.45 to 0.05 V and various electrolyte concentrations from 0.1 to 2.0 M, whereas AB and PSF were found to show symmetrical decreases in FEs with the applied potential

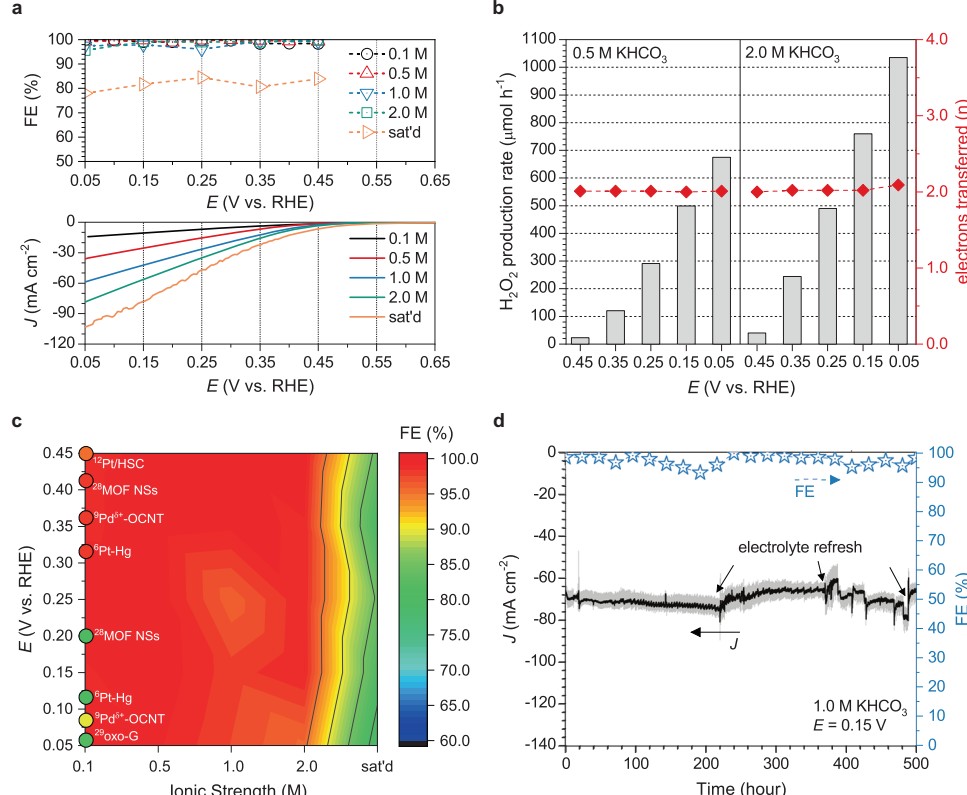

**Fig. 3 | 2e-ORR activities measured in a H-type cell.** The $H_2O_2$ yields were identified by the titration method after CA electrolysis. **a** ORR polarization profiles of D-PSFZ at the various solution concentrations of 0.1 to 2.0 M and saturated $KHCO_3$ solution (the lower part) and the corresponding FEs determined by the titration method (the upper part). **b** $H_2O_2$ production rates observed from CA tests in 0.5 M or 2.0 M $KHCO_3$ and corresponding the number of electrons transferred (n). **c** The color-mapped contour plot of FEs toward 2e-ORR for D-PSFZ as a function of potentials and solution concentrations. **d** The 2e-ORR stability profile tested in 1.0 M $KHCO_3$ at 0.15 V vs. RHE. The electrolyte was refreshed after 220, 370, and 480 h.

and electrolyte concentration (Supplementary Fig. 18). Figure 3c also shows that those state-of-the-art 2e-ORR catalysts (i.e., Pt/HSC[12], Pd[8+]-OCNT[9], MOF NSs[28]) reported recently exhibited FEs around 95% at the potentials over 0.30–0.60 V. Within a highly polarized potential range (i.e., 0.05–0.20 V), however, these $\Delta G_{HOO*}$-optimized electrocatalysts still showed a decrease in FEs, as represented by the colored circles, indicating that D-PSFZ even outperformed these reported precious-metal based electrocatalysts over a broad range of the potentials and electrolyte concentrations. As summarized in Supplementary Table 30, the D-PSFZ demonstrated superior performance compared to recent publications and exhibited highly selective 2e-ORR activities in a wide range of conditions (i.e., ionic strengths, overpotentials), making it well-suited for high current operations.

To identify the electrochemical stability of D-PSFZ, the CA profile was observed in 1.0 M $KHCO_3$ at the high $\eta$ region (i.e., 0.15 V) using a circulating pump and electrolyte reservoir (See Methods for details). As shown in Fig. 3d, D-PSFZ exhibited a stable current density of ~70 mA $cm^{-2}$ over 500 h, maintaining FEs around 98%. This clearly demonstrates that D-PSFZ can perform efficient and durable catalytic activities even at high current and high ionic strength conditions for scalable $H_2O_2$ electrosynthesis. Furthermore, we identified the 2e-ORR activities in a high ionic strength condition (i.e., 2 M $KHCO_3$) for the $\delta$-controlled D-PSFZs with different oxygen vacancy (Ov) contents prepared via quenching or reducing method (denoted Q-PSFZ and R-PSFZ, respectively) as described in Methods. Different Ov values of $\delta = 0.10$–0.11 and $\delta = 0.19$–0.20 were determined from iodometric titrations for Q-PSFZ and R-PSFZ, respectively, with no change in the crystal structure (Supplementary Figs. 19 and 20). As shown in Supplementary Fig. 21, however, the less oxygen-deficient sample, Q-PSFZ, presented a typical 2e-ORR profile showing a decrease in FE with

increasing $\eta$, as is the case with PSF (Supplementary Fig. 18). In contrast, the Ov-rich R-PSFZ showed an opposite trend with high FEs in the high $\eta$ region (Supplementary Fig. 21). These results provide clear evidence for the role of Ov in regulating the electrocatalytic performance for scalable production of $H_2O_2$ under high $\eta$ & ionic strength conditions. The CA profiles for the $\delta$-controlled R-PSFZ, Q-PSFZ, and D-PSFZs are shown in Supplementary Figs. 22, 23, and 16, respectively, along with the related data in Supplementary Tables 10–14. The 2e-ORR activities of D-PSFZ in 0.1 M KOH showed FEs > 99% over the entire ORR range (Supplementary Fig. 24, Supplementary Table 15). The $H_2O_2$ yields were also confirmed by UV-vis spectroscopy, using $TiOSO_4$ as a coloring-reagent and calibration profiles in Supplementary Fig. 25 (see, Method). Supplementary Figs. 26 and 27 and the associated insets indicate that the $H_2O_2$ yields determined by the titration are consistent with those by the UV-vis methods. The above discussions confirm the excellent selective 2e-ORR pathway towards $H_2O_2$ synthesis for D-PSFZ with highly efficient and stable FEs around 99% over a broad ORR potential range (i.e., 0.45–0.05 V) at various electrolyte concentrations (i.e., 0.1–2.0 M $KHCO_3$) over 500 h.

## 2e-WOR activities

We further investigated 2e-WOR activities for the D-PSFZ in various $KHCO_3$ concentrations from 0.1 to 2.0 M. Figure 4a shows the CV polarization curves (the lower part) and the $H_2O_2$ selectivity (the upper part) for D-PSFZ determined by the titration method after the CA measurements (i.e., 2.05–2.35 V, every 0.05 volts) for 30 min (see also, Supplementary Fig. 28 and Supplementary Tables 16–19). As shown in Fig. 4a and Supplementary Fig. 28c, D-PSFZ displayed a maximum FE ($FE_{max}$) of 52.7% at 2.10 V in 0.1 M $KHCO_3$, which decreased at higher potentials (i.e., FE = 36.6% at 2.35 V). At 0.5 M $KHCO_3$, the higher

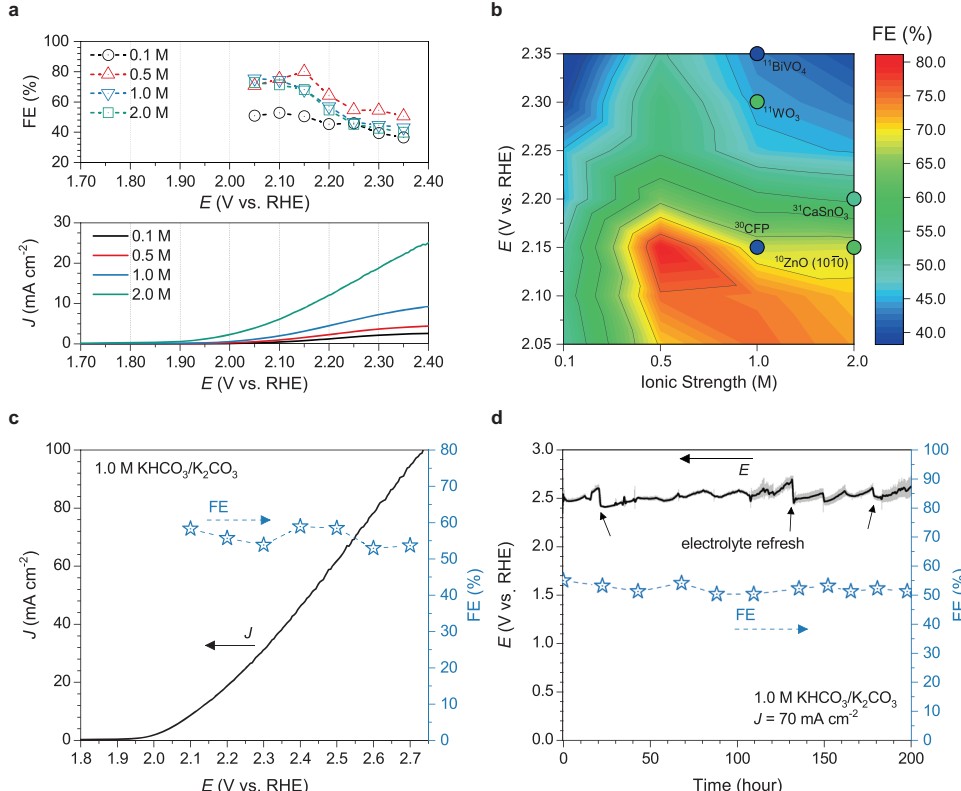

**Fig. 4 | 2e-WOR activities measured in a H-type cell.** The $H_2O_2$ yields were identified by the titration method after CA electrolysis. **a** WOR polarization profiles of D-PSFZ at the various solution concentrations of 0.1–2.0 M and saturated KHCO3 solution (the lower part) and the corresponding FEs determined by the titration method (the upper part). **b** The color-mapped contour plot of FEs toward 2e-WOR for D-PSFZ as a function of potentials and solution concentrations. **c** A WOR polarization profile of D-PSFZ in 1.0 M KHCO3/K2CO3 (pH = 10) solution and the corresponding FEs determined by the titration method. **d** a 2e-WOR stability profile tested in 1.0 M KHCO3/K2CO3 at 70 mA cm$^{-2}$. The electrolyte was refreshed after 20, 130, and 175 h.

current density profiles were found around 1–4 mA cm$^{-2}$ at the potential range from 2.05 to 2.35 V (Supplementary Fig. 28b). As shown in Fig. 4a (the upper part) and Supplementary Fig. 28d, D-PSFZ exhibited a maximum FE$_{max}$ up to 80.0% at 2.15 V. When D-PSFZ was tested at a higher KHCO3 concentration (i.e., 1.0 and 2.0 M KHCO3), however, much higher current densities, ranging from 10 to 20 mA cm$^{-2}$, were observed (Supplementary Fig. 28e, f and Fig. 4a), leading to a slight decrease in the maximum FE to 75.4% and 72.8% for 1.0 and 2.0 M KHCO3 solutions, respectively, at 2.05 V *vs*. RHE (Supplementary Fig. 28g, h and Fig. 4a).

Figure 4b displays the color-mapped contour plot of FEs for D-PSFZ toward 2e-WOR as a function of the potential and KHCO3 concentration, along with those state-of-the-art catalysts reported recently[10,11,30,31]. Compared with the reported FE of around 70% for (10$\bar{1}$0) ZnO (2 M KHCO3)[10] at 3.0 V *vs*. RHE, 70% for BiVO4 (1 M NaHCO3)[11] at 3.1 V, and 76% for CaSnO3 (2 M KHCO3)[31] at 3.2 V, D-PSFZ showed a notably higher FE of around 80% at significantly lower working potential of 2.15 V in 0.5 M KHCO3. In view of the fact that the decomposition of metal oxides could occur at high potentials (>3.0 V *vs*. RHE), the low and moderate working potentials around 2.0 V for the high FE (~80%) observed for D-PSFZ is especially attractive for durable practical electrolysis[19,20]. At potentials around 2.0 V, D-PSFZ also exhibited about 20% higher FEs than the corresponding values for the recently-reported 2e-WOR catalysts (Fig. 4b), including ZnO (10$\bar{1}$0) (47% FE at 2.15 V)[10], CFP (25% FE at 2.15 V)[30], CaSnO3 (35% FE at 2.20 V)[31], WO3 (45% FE at 2.30 V)[11], and BiVO4 (23% FE at 2.35 V)[11]. Since carbonate ion ($CO_3^{2-}$) can contribute to yield more $H_2O_2$ during oxidation reactions as promoters[11,31], we also have further examined WOR profiles in 1.0 and 2.0 M KHCO3/K2CO3 (pH ~ 10.0) (Fig. 4c and Supplementary Fig. 29). Because the second proton dissociation of bicarbonate

(p$K$a$_2$ = 10.32, $HCO_3^- \rightarrow H^+ + CO_3^{2-}$) can occur around pH = 9.0 and higher (Supplementary Fig. 30), thus, the weak-alkaline conditions are favored. In 1.0 and 2.0 M KHCO3 conditions (Fig. 4b and Supplementary Fig. 28), the selectivity maintained high (FEs of more than 55%) above 2.3 V whereas it decreased (FEs ~ 40%) at high $\eta$ region (e.g., 2.3 V). In 1.0 M KHCO3/K2CO3, D-PSFZ exhibited the FE$_{max}$ up to 60% at 2.5 V (i.e., $\eta$ = 0.74 V) performing high current operations, i.e., ~70 mA cm$^{-2}$ (Fig. 4c), indicating that D-PSFZ can be utilized for bifunctional $H_2O_2$ electrosynthesis at high ionic strength conditions for scalable production of $H_2O_2$. Figure 4d presents the electrochemical stability of D-PSFZ toward WOR at the high current density of 70 mA cm$^{-2}$. The D-PSFZ exhibited a stable potential profile of ~2.5 V *vs*. RHE over 200 h, maintaining the FEs around 52%. Notably, these WOR results with a high current density (>50 mA cm$^{-2}$) at a low overpotential ($\eta$ < 1.0 V) are much beneficial than those for the recently reported electrocatalysts, as summarized in Supplementary Table 31. The 2e-ORR activities of D-PSFZ in 1.0 and 2.0 M KHCO3/K2CO3 can be found in Supplementary Fig. 31, Supplementary Figs. 32 and 33 and the associated insets confirm that the $H_2O_2$ yields determined by the titration are consistent with those by the UV-vis methods.

## DFT calculations for mechanistic studies

The density functional theory (DFT) calculations were employed to investigate the intrinsic activity of D-PSFZ towards $H_2O_2$ generation. Simulation details and structural models to identify the most stable RP framework can be found in Methods and Supplementary Figs. 34–37. Figure 5a shows the reaction mechanism for $H_2O_2$ production on the surface of the Fe-O end facet and the reaction intermediates during ORR (red arrow) and WOR (black arrow). This indicates that the efficiency of ORR and WOR could be thermodynamically evaluated by the

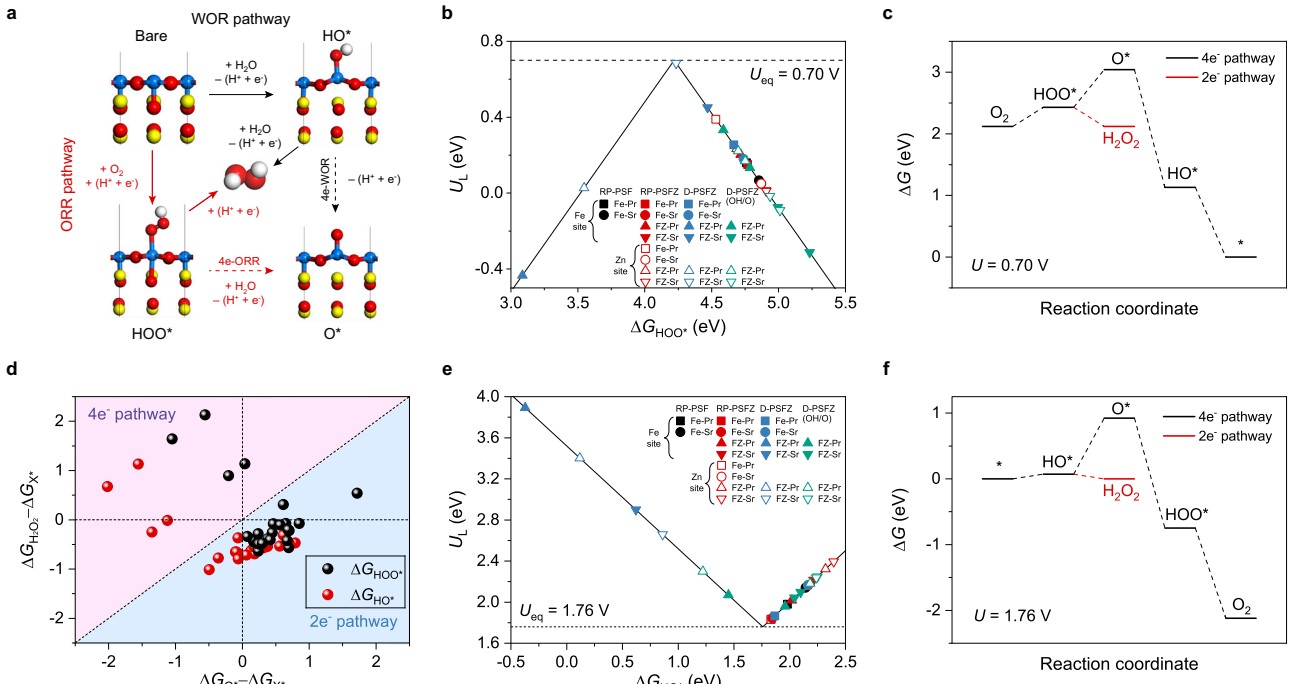

**Fig. 5 | DFT calculations of D-PSFZ toward 2e-ORR and 2e-WOR. a** Schematic of $H_2O_2$ generation mechanism on PSF surface through oxygen reduction (red arrow) and water oxidation reaction (black arrow). Blue, yellow, red, and white balls represent Fe, Pr, O, and H atoms, respectively. **b, e** Volcano relation between limiting potential and reaction intermediate for ORR and WOR. Each model was named by the information of B-site ions in the top layer and A-site ions in the second layer;

Fe-Pr, Fe-Sr, FZ-Pr, FZ-Sr ('FZ' represents Fe/Zn layer). **c, f** The reaction mechanism on the best active site (i.e., high selectivity and activity) for ORR and WOR. **d** Energy differences between reaction intermediates at equilibrium potential. The blue (or red) region represents the active sites for the thermodynamically driven 2e⁻ (or 4e⁻) reaction pathway.

energy states of single reaction intermediates, such as HOO* for ORR and HO* for WOR. $\Delta G_{HOO*}$ and $\Delta G_{HO*}$ are utilized as parameters for predicting the catalytic activities toward 2e-ORR and 2e-WOR, respectively. In principle, to efficiently produce $H_2O_2$, the limiting potential for each reaction (i.e., energy barrier at equilibrium potential) should be close to 0 eV. In the case of PSF where Zn is not located in the B-site, the limiting potential was evaluated as 0.16/0.07 eV and 1.98/2.14 eV for ORR and WOR, respectively (Fig. 5b, e). These activity results were positively broadened after Zn-doping on the B-site and the introduction of O$v$ by varying the surface configurations. Thus, both Zn and Fe atoms serve as an active site for both ORR and WOR. This implies that the Zn dopant and the associated formation of O$v$ can enhance the catalytic activity for both nearby atoms of Fe and Zn on the surface. The optimal limiting potentials for ORR and WOR were 0.39 and 1.83 eV, respectively.

Although $\Delta G_{HOO*}$ and $\Delta G_{HO*}$ are predicting indicators for 2e-ORR and 2e-WOR catalytic activities, they account for the energy barrier required at the equilibrium potential (i.e., limiting potential, $U_L$) to initiate the reaction, but are not directly related to reaction selectivity (i.e., FEs). In view of the turnover track of these ORR and WOR steps, the competitive reaction step against $H_2O_2$ generation is the formation of O* intermediate (Fig. 5a) for both reaction pathways. Thus, for the highly selective surface towards $H_2O_2$ generation, the formation of reaction intermediate should be favorable to $H_2O_2$ rather than O*. Therefore, although $\Delta G_{HOO*}$ and $\Delta G_{HO*}$ can be used as descriptors to describe ORR or WOR, $\Delta G_{O*}$ is more proper to predict the favorability of $H_2O_2$ against O* formation in both ORR and WOR. In this regard, we introduce a descriptor $\Delta G_{H_2O_2} - \Delta G_{X*}$ (X = HOO or HO) as a function of another descriptor $\Delta G_{O*} - \Delta G_{X*}$ to build a mechanistic map for the reaction selectivity at the exposed surfaces of D-PSFZ (Fig. 5d). As shown in Fig. 5d, few active sites locate in the 4e⁻ pathway region (red area), where the O* state could be thermodynamically favorable. However, most of these sites have much lower energy states than

those for potential-determining reaction intermediates, i.e., $\Delta G_{O*} - \Delta G_{X*}$ (X = HOO or HO) < 0, suggesting that the dominant reaction pathways are hard to be determined on these facets. For the majority cases of the surface, however, it was found that $\Delta G_{O*} - \Delta G_{X*}$ values are greater than $\Delta G_{H_2O_2} - \Delta G_{X*}$ values (Fig. 5c, d, f), indicating that the binding state of O* is significantly unfavorable and unstable than the $H_2O_2$ formation pathways. In contrast to the state-of-the-art single-atom catalysts[32,33], D-PSFZ offered the active sites with high selectivity towards $H_2O_2$ production which was driven by reaction thermodynamics. Based on our experimental evidence, we found that $\Delta G_{O*}$ is a more decisive parameter to indicate selective FEs at high ionic strengths or high $\eta$ conditions. Therefore, we confirmed that the surface configurations of D-PSFZ were varied by the introduction of Zn dopant and O$v$ in the lattice, leading to highly active surfaces towards $H_2O_2$ generation. In addition, we discovered that $\Delta G_{O*}$ can be used to explain that the thermodynamically unstable energy states of O* can lead to the highly selective $H_2O_2$ electrosynthesis during a turnover track of ORR and WOR for D-PSFZ.

## D-PSFZ‖D-PSFZ Cell for Overall $H_2O_2$ Electrosynthesis

In this study, we have demonstrated the full cell configuration for overall $H_2O_2$ electrosynthesis using D-PSFZ as both cathode and anode (denoted as D-PSFZ‖D-PSFZ) without using a membrane to separate the electrodes. Other attempts to combine different catalyst electrodes are summarized in Supplementary Table 32. Typically, a membrane is utilized to separate two electrodes to avoid the decomposition of $H_2O_2$ at the anode, but this membrane-free design is beneficial for overcoming the low ionic conductivity and eliminating additional resistance, cell fabrication complexity and/or cost associated with the membrane usage (solution resistance upon various electrolyte conditions can be found at Supplementary Fig. 38)[34]. For the full-cell testing, we placed both electrodes in a beaker with an O₂-sat'd $KHCO_3$ or $KHCO_3/K_2CO_3$ electrolytes in a three-electrode configuration using an

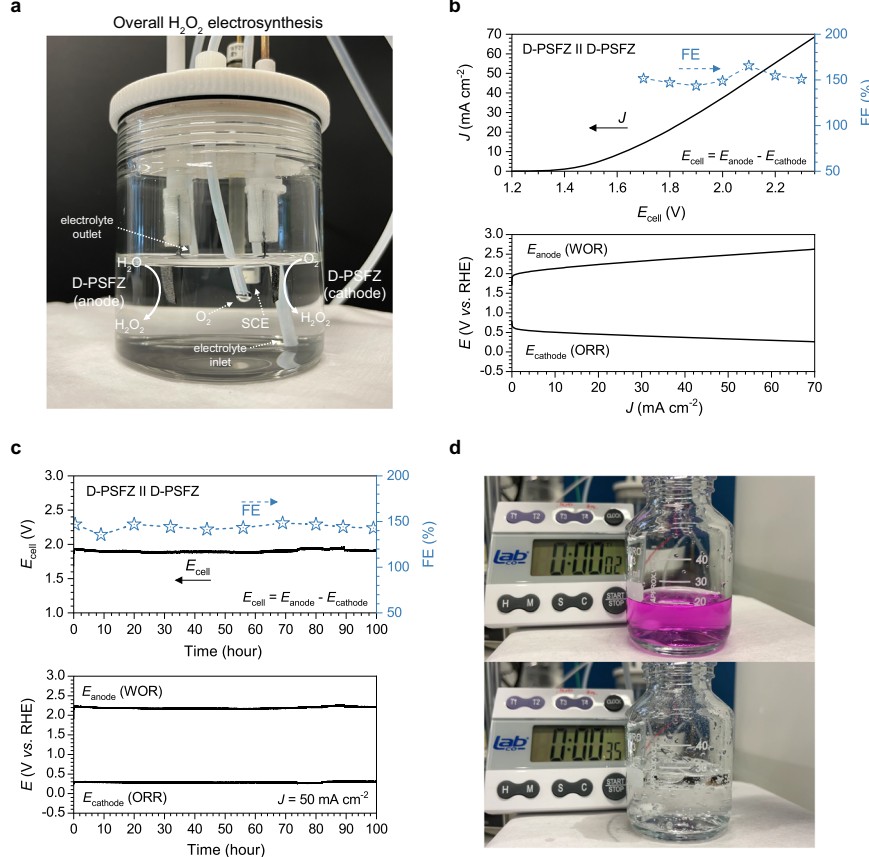

**Fig. 6 | Overall H₂O₂ electrolysis performance. a** Photograph of the membrane-free overall H₂O₂ electrosynthesis unit in a three-electrode configuration using the D-PSFZ as both cathode and anode (denoted as D-PSFZ‖D-PSFZ) with a SCE reference electrode. **b** Polarization I-V profiles of D-PSFZ‖D-PSFZ measured in O₂-sat'd 1.0 M KHCO₃/K₂CO₃ and the corresponding FE profiles obtained by the titration method. **c** The stability profile for D-PSFZ‖D-PSFZ full cell ($E_{cell}$) measured in O₂-sat'd 2.0 M KHCO₃/K₂CO₃ at 50 mA cm⁻². The below parts show the anode ($E_{anode}$) and cathode ($E_{cathode}$) potentials. **d** Photograph of the in-situ dye decomposition experiment (2.5 μmol of permanganate) by the electrolyte outlet of the stability testing near the 80-h moment.

SCE reference electrode (Supplementary Figs. 39 and 6a). The total FE from the electrolysis was defined as the sum of the cathodic FE and the anodic FE, so the theoretical maximum is 200%. Figure 6b shows the ORR and WOR polarization curves, denoted $E_{cathode}$ and $E_{anode}$, respectively (the lower part), measured during the overall H₂O₂ synthesis and the obtained cell potentials ($E_{cell} = E_{anode} − E_{cathode}$) for D-PSFZ‖D-PSFZ (the upper part). The initiation voltage for the overall H₂O₂ electrosynthesis was found to be near 1.40 V, which requires a small $\eta$ of 0.3 V since the equilibrium potential is 1.08 V as follows: O₂ + 2H₂O ⇌ 2H₂O₂, $E^o$ = 1.08 V. As shown in Fig. 6b, we observed a well-balanced current profile between the cathodic and anodic currents in the 1.0 M KHCO₃/K₂CO₃ condition. Notably, D-PSFZ‖D-PSFZ exhibited a sharply increasing current density profile (i.e., exceeding 50 mA cm⁻² around 2.1 V) with increasing the potential (Fig. 6b) owing to its concurrently effective ORR and WOR kinetics.

To evaluate the FEs during the overall H₂O₂ electrosynthesis, we investigated the CA profiles of D-PSFZ‖D-PSFZ at the cell potential of 1.70–2.30 V (Supplementary Fig. 40) and the corresponding FEs were calculated by the titration and UV-vis methods. The titration and UV-vis data are listed in Supplementary Fig. 41, Supplementary Tables 24–26. As shown in the CA profile (Supplementary Fig. 40), the D-PSFZ‖D-PSFZ shows plateau-like current density profiles without a trend of increasing/decreasing current density, suggesting that the H₂O₂ product does not involve secondary reactions. Interestingly, the corresponding FEs were all identified above 150% in the cell potential range of 1.70–2.30 V and especially the FE_max was recorded to be 163.0% with ~50 mA cm⁻² operation (Fig. 6b, the upper part) in the 1.0 M KHCO₃/

K₂CO₃ electrolyte. Similarly, in the KHCO₃ conditions, the FE_max was found as 163.9% and 161.6% in the 0.5 M and 1.0 M KHCO₃ solution (Supplementary Fig. 39) at the cell potential of 1.80 V. These FEs tended to slightly decrease at higher potentials, but all the FEs were observed as high as 150%.

To investigate the electrochemical stability and durability of the D-PSFZ catalyst, we measured its chronopotentiometric profiles in the 2.0 M KHCO₃/K₂CO₃ electrolyte at a current density of 50 mA cm⁻² as shown in Fig. 6c. The D-PSFZ‖D-PSFZ exhibited an excellent electrochemical stability with a negligible fluctuation in performance at a high current density of 50 mA cm⁻² near 1.95 V over 100 h, maintaining FEs around 150%, suggesting that D-PSFZ can exhibit efficient bifunctional activities to generate H₂O₂ from 2e-ORR and 2e-WOR at high $\eta$ regions & ionic strength conditions for practical applications. Furthermore, we proceeded the in-situ dye decomposition experiment using the outlet electrolyte solution from the stability test around 80 h (Supplementary Movie 1 and Fig. 6d). We found that the dye decomposed within ~20 s corresponding to a FE of ~141% that is similar to the accurate FE of 146.9% (See Supplementary Note 1). To identify the structural stability of the D-PSFZ, we recorded XRD profiles for the electrodes after the stability testing. As shown in Supplementary Fig. 42, both the cathode and the anode exhibited the distinctive RP perovskite structure peaks without secondary phases and no noticeable change in the XRD profiles, suggesting a good structural stability toward the overall H₂O₂ electrosynthesis. Furthermore, Supplementary Fig. 43 presents the SEM images of the D-PSFZ electrodes before and after testing, indicating no noticeable clogging or electrochemical/physical damage.

High-resolution XPS spectra and the corresponding depth profiles (Supplementary Figs. 44–47) further demonstrate the stable chemical state for the D-PSFZ electrodes. Some subtle changes in the XPS Sr 3$d$ and O 1$s$ regions were observed with care, arising from the absorption of water and/or oxygen during the electrolysis[35]. HR-TEM images of the D-PSFZ anode (Supplementary Fig. 48a, b) and cathode (Supplementary Fig. 48d, e) show high-crystalline structure of RP perovskites, as confirmed from the XRD analyses (Supplementary Fig. 42). Notably, the inset images in Supplementary Fig. 48b (anode) and Supplementary Fig. 48e (cathode) display the enlarged lattice of the boxed areas, clearly revealing the distinctive (311) or (100) planes of the RP perovskite oxides, as illustrated in Fig. 48c and f, respectively. Clearly, therefore, D-PSFZ showed efficient bifunctional activities toward both 2e-ORR and 2e-WOR with a high selectivity and long-term stability, enabling the efficient overall $H_2O_2$ synthesis in a membrane-free configuration by suppressing various competitive reactions, such as 4e-ORR, 4e-WOR, and $H_2O_2$ oxidation.

In summary, we have developed a new class of oxygen-deficient perovskite oxides (i.e., D-PSFZ) rich of oxygen vacancies (O$v$) for highly selective $H_2O_2$ generation with FEs over 99% across the entire ORR potential region (0.05 to 0.45 V $vs$. RHE) in various electrolyte conditions (0.1 to 2.0 M $KHCO_3$) with an excellent stability over 500 h, significant for scalable $H_2O_2$ electrosynthesis. Moreover, D-PSFZ exhibited the notably high 2e-WOR selectivity of 80.0% at the significantly low potential of 2.15 V, whereas the recent 2e-WOR catalysts reported similar FEs around 3.0 V - the higher potential could cause possible oxide corrosions. By using the efficient bifunctional D-PSFZ as both cathode and anode in a membrane-free overall $H_2O_2$ electrolysis, we demonstrated a record-high FE of 163.0% for the D-PSFZ||D-PSFZ cell at a current density of 50 mA cm$^{-2}$. Our combined experimental study and DFT calculations revealed that the formation of O$v$ induced by doping of D-PSFZ with divalent ions (i.e., $Zn^{2+}$) can modify the electronic structures of nearby B-site cations to strengthen (but not too strong) their binding with the HOO* intermediate and introduce an energy state of O* desirable for 2e pathways, and that $\Delta G_{O^*}$ (rather than the $\Delta G_{HOO^*}$ parameter) is a more decisive parameter to indicate selective FEs. This work represents a breakthrough in not only fundamental understanding of the ORR and WOR bifunctional activities for perovskite oxides and other related catalysts but also large-scale, cost-effective, efficient, and clean electrochemical production of $H_2O_2$ of practical significance.

## Methods

### Synthesis of perovskite oxides
$Pr_{1.0}Sr_{1.0}Fe_{1.0-x}Zn_xO_{4-\delta}$ ($x = 0$, and 0.25) perovskite oxides were synthesized by typical sol-gel process and named as PSF, D-PSFZ, respectively. To confirm the structural stability, $Pr_{1.0}Sr_{1.0}Fe_{1.0-y}Zn_yO_{4-\delta}$ ($y = 0.50$, and 0.75) oxides were also prepared. Stoichiometric amount of metal nitrate precursors and citric acid were dissolved in deionized water to form an aqueous solution. After the nitrate precursors were completely dissolved, an appropriate amount of polyethylene glycol ($M_w$ ~400) was added. After a viscous resin was formed, the solution was heated at 300 °C. Then, the resulting powder was pre-calcined at 650 °C of air for 4 h and calcined at 1150 °C of air for 4 h. In this work, all chemicals were purchased from Sigma-Aldrich unless specific notes were described, and used as received without further purification. The $\delta$-controlled D-PSFZs with different $\delta$ values were prepared by the post-treatment process of quenching or reducing method (denoted Q-PSFZ and R-PSFZ, respectively). Q-PSFZ was prepared by annealing D-PSFZ at 1000 °C in air for 1 h, followed by rapidly placing them into liquid $N_2$. R-PSFZ was prepared by annealing D-PSFZ at 750 °C under humidified $H_2$ for 2 h.

### Physical characterizations
X-ray diffraction (XRD) patterns were investigated using powder XRD instrument (Rigaku diffractometer, MiniFlex, Cu Kα radiation) with a scan rate of 0.5° min$^{-1}$. The XRD pattern and lattice parameters were analyzed by Rietveld refinement using FullProf software. The microstructure of the catalysts was examined by scanning electron microscopy (SEM, Thermo Fisher Scientific). The transmission electron microscopy (TEM) and scanning TEM (STEM) images were obtained by a high resolution-TEM (JEOL, JEM-2100F). Brunauer–Emmett–Teller (BET) surface area analysis and Barrett–Joyner–Halenda (BJH) pore size and volume analysis were conducted by a surface area & porosity analyzer (Micromeritics, TriStar II). Thermogravimetric analysis (TGA) profiles were obtained by a thermo-microbalance (NETZSCH, TG 209 F3 Tarsus). X-Ray photoelectron spectroscopy (XPS) analysis was conducted by Thermo Scientific ESCALAB250Xi (monochromatic Al Kα ~1486.68 eV) and the XPS profiles were referenced by calibrating the binding energy of carbon 1$s$, i.e., $C_{1s}$, at 284.8 eV. The depth profiles of the XPS were obtained with the same equipment using pass energy of 100 eV for survey scans, or 50 eV for depth profiling region scans. Oxygen non-stoichiometry was identified via iodometry titration. Oxygen contents can be calculated from the mean oxidation state of the B-site cations in the perovskite oxides that evaluated from iodometry method. The excessive amount of potassium iodide (KI) around 1.0 g was dissolved in $N_2$-saturated HCl solution to prevent the oxidation of iodide ions by air. And approximately 100 mg of perovskite oxides were dissolved into the solution to form brown colored $I_3^-$(aq) based on the following equilibrium: $I_3^- + 2e^- \rightleftharpoons 3I^-$. Then the solution was titrated by 0.05 N (normal concentration) $Na_2S_2O_3$ aqueous solution (Sigma-aldrich, Fluka Analytical) to calculate the amount of transferred charge based on the following equation: $I_3^-$ (brown color) $+ 2S_2O_3^{2-} \rightarrow S_4O_6^{2-} + 3I^-$ (colorless).

### Rotating ring disk electrode (RRDE) testing
Half-cell measurements were conducted in typical three-electrode configuration (Pine Research MSR Rotator) using a graphite rod and a saturated calomel electrode (SCE) (saturated KCl filled) as the counter and reference electrode, respectively. Electrochemical tests were carried out using a computer-controlled bipotentiostat (Pine Research, WaveDriver 40). $iR$ drops were not compensated for all data set in this work. Catalysts were prepared in the form of an ink by dispersing 19 mg of the catalyst and 1 mg of the acetylene carbon black as a conducting additive in 1 mL of a binder solution (45:45:10 = ethanol:isopropyl alcohol:5 wt.% Nafion solution (Sigma-Aldrich), volumetric ratio) followed by a tip sonication process. Then, electrochemical testing was investigated in $O_2$-saturated aqueous solution at a scan rate of 10 mV s$^{-1}$ by drop-coating 5 μL of each catalyst ink onto glassy carbon disk electrode. A calibration in reversible hydrogen electrode (RHE) was experimentally determined at a scan rate of 1 mV s$^{-1}$ in $H_2$-saturated aqueous solution, where platinum wire was used as the working and counter electrode and SCE as the reference electrode. The number of electrons transferred ($n$) and peroxide yield (%) were calculated by the followed equations:

$$n = 4 \times \frac{I_d}{I_d + I_r/N} \tag{1}$$

$$H_2O_2(\%) = 200 \times \frac{I_r/N}{I_d + I_r/N} \tag{2}$$

where $I_d$ and $I_r$ are the disk current and ring current, respectively, and $N$ is the collection efficiency. The Faradaic efficiency (FE) was determined as follows:

$$FE(\%) = \frac{\text{charge used for 2e reaction}/2(C)}{\text{charge used for 2e reaction}/2 + \text{charge used for 4e reaction}/4(C)} \times 100(\%) \tag{3}$$

### H-cell electrochemical testing and $H_2O_2$ quantification methods
The electrochemical testing to identify the accumulated amount of hydrogen peroxide was conducted by typical H-type cell configuration

by using a Nafion membrane (Nafion 211, FuelCell Store) as a separator. For 2e-ORR measurement, the cathode was prepared by drop-coating the catalyst ink prepared for RRDE testing onto the carbon cloth (CeTech W1S1010, FuelCell Store) with a loading density of 2 mg cm$^{-2}$. For 2e-WOR testing, the anode was via drop-coating the catalyst ink prepared by dispersing 19 mg of the catalyst and 1 mg of the carbon black in 1 mL of solution (50:50 = ethanol:isopropyl alcohol, volumetric ratio) onto the carbon cloth (AvCarb P50, FuelCell Store) with a loading density of 2 mg cm$^{-2}$ followed by a drop-coating PTFE or PVDF solution on the electrode. For the stability test of ORR, WOR, and full-cell, the catalyst ink was drop coated onto the carbon cloth with a loading density of 2 mg cm$^{-2}$ and 50 uL of PVDF solution (0.25 wt.% in acetone) was drop coated. For stability tests, we supplied the electrolyte liquid using a circulation pump and the electrolyte reservoir into the cell, the reacted electrolyte was pumped out and collected to chemically decompose the $H_2O_2$ product using $MnO_2$ or graphite catalysts. The corresponding $H_2O_2$-removed electrolyte was supplied to the electrolyte reservoir every 20 h. We refreshed the electrolyte with totally new solution as described in the Figs. 3d and 4d. To identify the peroxide yields toward 2e-ORR and 2e-WOR, we utilized 3 methods, such as 1. $H_2O_2$ test strips (Quantofix, MACHEREY-NAGEL), 2. Permanganate titration (using 5 mM $KMnO_4$ solution) and 3. Ultraviolet-visible (UV-vis) spectroscopy (using $TiOSO_4$ solution as a coloring additive) after accumulating $H_2O_2$ using chronoamperometric (CA) measurement for 10–15 min. The permanganate titration can be used to determine accurate amount of $H_2O_2$ yield compared to the $H_2O_2$ stripping and the UV–vis methods because the stripping method has difficulty to read the color and the UV–vis method cannot be used for the high concentration of $H_2O_2$ (the observance over 1.0 can be erroneous). The permanganate titration is based on the following equilibrium: $5H_2O_2 + 2MnO_4^-$ (violet color) $+ 6H^+ \rightleftharpoons 8H_2O + 5O_2 + 6Mn^{2+}$ (colorless). In this work, 2.0 M sulfuric acid was added as the $H^+$ source (e.g., 1 mL of 2.0 M $H_2SO_4$ + 0.5 mL of $H_2O_2$ accumulated electrolyte). Data sheets for the permanganate titration are available in Supplementary Tables 5–29. The $H_2O_2$ yield can be determined from the UV–vis method by addition of $TiOSO_4$ as a coloring agent based on the following equation: $H_2O_2 + Ti(IV)OSO_4 \rightleftharpoons [Ti(OH)_2(H_2O)(H_2O_2)]^{2+}$ (yellow color) when pH below 2. For UV-vis spectra measurements, 0.2 mL of 20 mM $TiOSO_4$ dissolved in 2.0 M $H_2SO_4$ solution was added in the $H_2O_2$ accumulated solution. The calibration profiles and details are available at Supplementary Fig. 25. Also, 2.0 M sulfuric acid was added as the $H^+$ source (e.g., 0.4 mL of 2.0 M $H_2SO_4$ + 0.4 mL of $H_2O_2$ accumulated KHCO$_3$ solution +0.2 mL of 20 mM $TiOSO_4$ in 1.0 M $H_2SO_4$). The FE determination during the stability testing can be found at Supplementary Note 1. The 1.0 M KHCO$_3$/K$_2$CO$_3$ electrolyte (total 1.0 M salt) indicates that the 1:1 (vol. ratio) mixture solution of 1.0 M KHCO$_3$ and 1.0 M K$_2$CO$_3$.

## Simulation details

Vienna ab initio simulation package (VASP) with the projector-augmented wave (PAW) method was used for spin-polarized density functional theory (DFT) calculations[36,37]. The electron exchange-correlations were described by generalized gradient approximation (GGA) with the Perdew−Burke−Ernzerhof (PBE) functional[38]. The GGA + $U$ scheme[39] was applied with the correction parameter of $U$ = 5.3 eV for Fe 3$d$ orbitals[40]. The energy cutoff was set to be 520 eV. The Monkhorst-Pack scheme[41] was used to sample the Brillouin zone of perovskite surface with $5 \times 5 \times 1$ $k$-points for actual spacing of 0.036 Å$^{-1}$. Moreover, VASPsol[42,43] was employed to correct the implicit solvation effect of water on the surfaces and reaction intermediates on them. All perovskite systems were optimized until the self-consistent forces reached 0.02 eV Å$^{-1}$ and the total energy was changed within 10$^{-6}$ eV per atom. The vacuum slab of ~20 Å was introduced along the $c$-axis for surface system to avoid self-interactions.

The unit cell of Ruddlesden-popper phased PSF oxide (RP-PSF) was comprised of Pr$_{4.0}$Sr$_{4.0}$Fe$_{4.0}$O$_{16.0}$ while Zn ion was substituted on one of the B-site in the RP-PSF to model the RP-PSFZ system (i.e., Pr$_{4.0}$Sr$_{4.0}$Fe$_{3.0}$Zn$_{1.0}$O$_{16.0}$) (Supplementary Fig. 34). One of lattice oxygen was removed in the unit cell of RP-PSFZ to describe the defective system corresponding to the amount of oxygen vacancy as $\delta$ = 0.25 in Pr$_{1.0}$Sr$_{1.0}$Fe$_{0.75}$Zn$_{0.25}$O$_{4.0-\delta}$ (D-PSFZ). The energetically stable structure of defective system was used in the simulation (Supplementary Fig. 35). For the perovskite surface, symmetric and nonstoichiometric slab was introduced to minimize the dipole effect (Supplementary Fig. 36). On the D-PSFZ surface, surface Fe atoms in Fe perovskite layer were five-coordinated while surface Fe and Zn atoms in Fe/Zn perovskite layer were four-coordinated. Four-coordinated surface atoms could be healed by strong interaction between the surface and oxygen reaction intermediates (HO* or O*) during water oxidation and oxygen reduction reaction. Thus, the oxidized surfaces were additionally considered for the possible configuration (Supplementary Fig. 37).

## Data availability

All data supporting the findings of this study are available within the article, as well as the Supplementary Information file. All other data supporting the findings of the study are available from the corresponding author upon request.

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

## Acknowledgements

L.D. is grateful for the financial support by ARC (FL 190100126 and CE230100032). G.K. thanks for the financial support by CAS Pioneer Hundred Talents Program.

## Author contributions

C.K., G.K. and L.D. conceived the idea and directed research. C.K. carried out material synthesis, characterization, and activity testing. S.P., and S.K.K. carried out density functional theory calculations. C.K., S.P., Z.X. and L.D. wrote the paper. All authors contributed to analyzing results and discussion.

## Competing interests

The authors declare no competing interests.
