## [Peer Review File · Nature Communications]

REVIEWERS' COMMENTS

Reviewer #1 (Remarks to the Author):

The authors have now addressed most of my questions.

Reviewer #3 (Remarks to the Author):

The authors have addressed most of my previous comments. One minor issue left is the reference for U value used in the GGA+U calculation. The authors cited ref. 38 but that is for nickel oxide, not iron. Please double check. After that it can be published

Point-to-Point Responses to the Reviewers Comments (NCOMMS-23-19091B)

Reviewer #3 (Comments for the Author):

The authors have addressed most of my previous comments. One minor issue left is the reference for U value used in the GGA+U calculation. The authors cited ref. 38 but that is for nickel oxide, not iron. Please double check. After that it can be published.

Response: Thank you for the Reviewer's careful reading. Sorry, we have made a mistake in numbering the references. From our original manuscript (Method part: Page 23, line 19), we wrote the following statement: "The GGA+U scheme³⁸ was applied with the correction parameter of $U = 5.3$ eV for Fe-3d orbitals³⁸." And it should be rewritten "The GGA+U scheme³⁸ was applied with the correction parameter of $U = 5.3$ eV for Fe-3d orbitals³⁹." Ref. 39 in our original manuscript explains the Hubbard U parameter for the Fe element (5.3 eV) (See **Figure R1**). It was our mistake not to cite the reference properly, even though it was listed in the References section. Accordingly, we addressed the Reviewer's concern by making the necessary correction in our further revised manuscript. We are grateful for the Reviewer's careful comment and hope this further revised manuscript is acceptable for publication.

Figure R1. The screen captured image of Ref. 39 of our original manuscript, which explains the Hubbard U parameter for the Fe element (5.3 eV). See: <https://www.nature.com/articles/s41524-019-0199-7>

References (Part of our original manuscript)

39. Horton, M. K. *et al.* High-throughput prediction of the ground-state collinear magnetic order of inorganic materials using Density Functional Theory. *npj Comput. Mater.* **5**, 64 (2019).